# Web and phone-based COVID-19 syndromic surveillance in Canada: A cross-sectional study

Lauren Lapointe-Shaw[1,2]*, Benjamin Rader[3,4], Christina M. Astley[3,5], Jared B. Hawkins[3,5], Deepit Bhatia[1], William J. Schatten[6], Todd C. Lee[7], Jessica J. Liu[1,2], Noah M. Ivers[8,9], Nathan M. Stall[2,10], Effie Gournis[11], Ashleigh R. Tuite[12], David N. Fisman[2,12], Isaac I. Bogoch[1,2⦿], John S. Brownstein[3,5⦿]

1 Department of Medicine, University Health Network, Toronto, Canada, 2 Department of Medicine, University of Toronto, Toronto, Canada, 3 Computational Epidemiology Lab, Boston Children's Hospital, Boston, MA, United States of America, 4 Department of Epidemiology, Boston University, Boston, MA, United States of America, 5 Department of Pediatrics, Harvard Medical School, Boston, MA, United States of America, 6 Forum Research, Toronto, Canada, 7 Department of Medicine, McGill University Health Centre and Clinical Practice Assessment Unit, McGill University, Montreal, Canada, 8 Department of Family and Community Medicine, University of Toronto, Toronto, Canada, 9 Department of Family Medicine, Women's College Hospital, Toronto, Canada, 10 Department of Medicine, Sinai Health System, Toronto, Canada, 11 Toronto Public Health, Toronto, Canada, 12 Dalla Lana School of Public Health, University of Toronto, Toronto, Canada

⦿ These authors contributed equally to this work.
* lauren.lapointe.shaw@utoronto.ca

**Data Availability Statement:** The data used in this study is the property of the Angus Reid Institute, Forum and Mainstreet Research, and the Boston

## Abstract

### Background

Syndromic surveillance through web or phone-based polling has been used to track the course of infectious diseases worldwide. Our study objective was to describe the characteristics, symptoms, and self-reported testing rates of respondents in three different COVID-19 symptom surveys in Canada.

### Methods

This was a cross-sectional study using three distinct Canada-wide web-based surveys, and phone polling in Ontario. All three sources contained self-reported information on COVID-19 symptoms and testing. In addition to describing respondent characteristics, we examined symptom frequency and the testing rate among the symptomatic, as well as rates of symptoms and testing across respondent groups.

### Results

We found that over March- April 2020, 1.6% of respondents experienced a symptom on the day of their survey, 15% of Ontario households had a symptom in the previous week, and 44% of Canada-wide respondents had a symptom in the previous month. Across the three surveys, SARS-CoV-2-testing was reported in 2–9% of symptomatic responses. Women, younger and middle-aged adults (versus older adults) and Indigenous/First nations/Inuit/Métis were more likely to report at least one symptom, and visible minorities were more likely to report the combination of fever with cough or shortness of breath.

Children's Hospital. Data from the Forum and Mainstreet poll have been made available at https://doi.org/10.5683/SP2/YM8BCJ. Requests for access to Angus Reid or COVID Near You data should go to info@angusreid.org or john.brownstein@childrens.harvard.edu, respectively. Data from COVID Near You to be used for public health surveillance purposes can be requested at: https://www.atscale.com/covidnearyou/.

**Funding:** This research was supported by the University of Toronto, Department of Medicine COVID-19 Funding Opportunity (Lapointe-Shaw). WJ Schatten is a paid employee of Forum Research, which collected and supplied the data resulting from the Forum & Mainstreet poll. Forum Research was not involved in study design, collection of data from other sources, data analysis, decision to publish, or preparation of the manuscript. Forum Research provided no other financial support to any study team members or their institutions. The specific role of all authors are described in the "author contributions" section.

**Competing interests:** WJ Schatten is a paid employee of Forum Research. This commercial affiliation does not alter our adherence to PLOS ONE policies on sharing data and materials. II Bogoch has consulted to BlueDot, a social benefit corporation that tracks the spread of emerging infectious diseases. The remaining authors have no disclosures.

## Interpretation

The low rate of testing among those reporting symptoms suggests significant opportunity to expand testing among community-dwelling residents of Canada. Syndromic surveillance data can supplement public health reports and provide much-needed context to gauge the adequacy of SARS-CoV-2 testing rates.

## Introduction

While SARS-CoV-2 has rapidly spread globally, ascertaining its true incidence remains a challenge [1, 2]. This is because a large proportion of those infected (20–75%) are minimally symptomatic or asymptomatic [3, 4]. Further, in many regions only those with severe illness or identified as a priority group are tested, and thus eligible for laboratory test-based confirmation [5]. Until a rapid test is widely available or barriers to diagnostic testing in Canada are lowered, there will be a reliance on symptoms for early detection [1]. Yet, the range of presenting symptoms is broad, including generally common complaints (headache, fatigue) and more specific symptoms such as loss of smell or new onset chilblains [6–9].

Syndromic surveillance is a public health tool that has been used extensively to identify the beginning of seasonal influenza outbreaks in the United States [10–12] and Canada, and for other viral and bacterial diseases globally [13]. Where testing is incomplete, self-reported symptoms data is used to supplement confirmed case counts and estimate the true extent of disease [1]. The value of syndromic surveillance is higher when syndromes are illness-specific. However, because of the broad range of symptomatic presentations observed in SARS-CoV-2-infected individuals, a highly specific definition is likely to lack sensitivity and miss most people who would be eligible for testing [7]. Whereas grouping symptoms into clinical syndromes is likely to increase specificity, looking at the occurrence of any described symptom is the most sensitive way to measure all those who would be eligible for COVID-19 testing.

In Canada, phone and internet methods have been used to collect symptomatic and testing information from voluntary public participants. The primary objective of this study was to describe the characteristics, symptoms, and self-reported testing rates of respondents across three different COVID-19 symptom and testing surveys. The one phone and two internet-based polls we studied covered varied population subsets and timeframes.

## Methods

In this cross-sectional study we retrospectively analyzed existing phone and internet survey data. This study was approved by the Ethics Review Board of University Health Network, which waived the requirement for informed consent. The data were de-identified prior to sharing with our study team. The only remaining identifiers were age, gender, and the first three digits of a six-digit Canadian postal code [14].

### Data sources

Three data sources were used for this study. Survey response rates and relevant survey questions are in S1–S4 Tables.

The Angus Reid Institute COVID-19 symptom poll was administered online from April 1–6, 2020 to a randomly selected sample of Angus Reid Forum panel members (a group of over 50,000 Canadian residents who have volunteered to regularly fill out surveys in exchange for gift card or

sweepstake rewards) [15, 16]. Respondents were asked about symptoms during the previous month, and about SARS-CoV-2 testing. Respondents were not asked about test results.

COVID Near You (covidnearyou.org) is a web-based participatory health surveillance tool created by infectious disease epidemiologists at Boston Children's Hospital [17]. This team also created Flu Near You (flunearyou.org), a similar tool for influenza symptoms, which has been validated against clinical data sources and applied to predict influenza trends [10–12]. Between the Canadian launch on April 3rd and April 26th, there were over 420,000 responses. For individuals opting to include their phone number to be contacted for follow-up surveys (12% of responses) subsequent responses with the same age/sex/phone number were excluded (N = 3,511). Respondents were asked to report on present symptoms, and related healthcare encounters, testing, and results.

The Forum & Mainstreet Research poll on COVID-19 symptoms was administered by telephone and SMS (text) message to randomly selected households in Ontario in two waves: April 11–12 and April 18–19, 2020 [18, 19]. Datasets from both survey waves were combined; only the first survey was used for households that appeared in both waves (N = 158). Respondents were asked to report on new symptoms in the household over the previous week, about testing since the onset of symptoms, and test results.

### Measures

Symptoms of possible COVID-19 were defined as inclusive of any of the following, where information was consistently available (>50% of sample was exposed to the question): fever, fatigue, runny nose, cough, aches and pains, chills/night sweats, sore throat, diarrhea, headache, shortness of breath, nausea, and loss of taste or smell. We excluded sneezing and rash as these are not described symptoms of COVID-19. We also reported on the self-reported combination of fever with either cough or shortness of breath, a COVID-like illness definition used by the World Health Organization [20]. Where possible, demographic variables were categorized to facilitate qualitative comparison between surveys.

### Analysis

Due to considerable methodological differences across sources, results were analyzed separately. Where survey weights were included in sources (Angus Reid and Forum polls), we reported unweighted counts and weighted frequencies. As the COVID Near You team does not derive or use survey weights, we report unweighted counts and frequencies for results from this source. For Canada-wide data reported at the individual-level (Angus Reid Institute and COVID Near You surveys), we further reported the frequency of any symptom, the syndrome of fever with cough or shortness of breath [20], and testing across demographic groups. For data reported at the household level (Forum poll), we reported the frequency of symptoms, testing, and test results across household size and income groups. Testing for differences was done using Rao-Scott Chi-square tests for weighted results and Chi-square tests and Fisher exact tests (if small cells) for unweighted results, all at a two-tailed $p < 0.05$ significance threshold. The data were analyzed using SAS software, version 9.4 (SAS Institute Inc., Carey, NC).

### Results

#### Angus Reid Poll- Canada-wide, April 1–6, 2020

There were 4,240 respondents, their median age was 46.5 years (IQR 33–61), 52.0% (n = 2,152) were women, nearly half had completed some college or university (46.8%, n = 2,023), and 13.1% (n = 529) reported being a visible minority (Table 1). Completed testing was reported

**Table 1. Self-reported characteristics of respondents in each of the three data sources[a].**

| | Angus Reid Institute N = 4,240 Individuals | COVID Near You N = 409,207 Responses | Forum/Mainstreet N = 9,147 Ontario households |
|---|---|---|---|
| **Age group of respondent, n (%)** | | | |
| Under 35 years | 1,197 (28.3) | 114389 (28.0) | 1,288 (13.0) |
| 35–54 | 1,491 (34.6) | 195140 (47.7) | 2,854 (31.2) |
| 55–64 | 755 (17.9) | 64765 (15.8) | 2,119 (24.0) |
| 65–74 | 618 (14.8) | 29855 (7.3) | 1,798 (19.6) |
| 75+ years | 179 (4.4) | 5057 (1.2) | 1,088 (12.2) |
| **Gender of respondent, n (%)** | | | |
| Female | 2,152 (52.0) | 237,150 (58.0) | 4,931 (53.3) |
| Male | 2,066 (47.6) | 164,487 (40.2) | 4,044 (45.0) |
| Other/No response | 22 (0.4) | 7,570 (1.8) | 172 (1.7) |
| **Annual Household Income (CAD), n (%)[b]** | | | |
| Under 25,000 | 422 (9.7) | - | 842 (7.3)[b] |
| 25,000-<50,000 | 761 (17.5) | | 2,719 (24.4)[b] |
| 50,000-<100,000 | 1,296 (30.3) | | 1,937 (20.3)[b] |
| 100,000-<150,000 | 762 (18.3) | | 1,860 (28.4)[b] |
| 150,000-<200,000 | 312 (7.7) | | |
| >200,000 | 166 (4.1) | | |
| Don't know/rather not say | 521 (12.4) | | 1,789 (19.6)[b] |
| **Highest Level of Education of Respondent, n (%)** | | | |
| Secondary or less | 1,043 (25.1) | - | 1,829 (18.3) |
| Some college or university | 2,023 (46.8) | | 3,335 (34.7) |
| Completed undergraduate | 819 (19.4) | | 2,405 (27.6) |
| Post-graduate degree | 355 (8.8) | | 1,578 (19.4) |
| **Respondent is Indigenous/First Nations/Inuit/Métis, n (%)** | 321 (7.3) | - | - |
| **Respondent is a visible minority, n (%)** | 529 (13.1) | - | - |
| **Household size, n (%)** | | | |
| 1 | 693 (15.8) | - | 1,620 (23.9) |
| 2 | 1,637 (38.1) | | 3,362 (34.5) |
| 3 | 790 (19.0) | | 1,526 (16.0) |
| 4 | 715 (17.3) | | 1,525 (15.3) |
| 5+ | 405 (9.8) | | 1,114 (10.4) |
| **Province, n (%)** | | | |
| Alberta | 422 (11.2) | 55,257 (13.5) | - |
| BC | 788 (13.1) | 70,634 (17.3) | |
| Manitoba | 259 (3.5) | 15,239 (3.7) | |
| New Brunswick | 81 (1.8) | 5,765 (1.4) | |
| Newfoundland/Labrador | 73 (1.8) | 1,786 (0.4) | |
| Nova Scotia | 147 (3.4) | 13,220 (3.2) | |
| Ontario | 1,200 (37.7) | 214,300 (52.4) | |
| PEI | 9 (0.2) | 571 (0.1) | |
| Quebec | 1,010 (24.1) | 20,344 (5.0) | |
| Saskatchewan | 251 (3.1) | 11,777 (2.9) | |
| Northwest Territories | - | 102 (0.0) | |
| Yukon | - | 176 (0.0) | |
| Nunavut | - | 21 (0.0) | |

[a] Cells <6 have been suppressed (denoted with a "-").

[b] The household income categories for the Forum/Mainstreet poll are: Under 20,000, 20,000–60,000, 60,000–100,000, >100,000, and "rather not say".

by 1.3% (n = 53), while 2.1% (n = 93) were not able to get tested, and 30.7% (n = 1,338) completed a COVID-19 self-assessment through a government website or app.

Over the previous month n = 1,863 (43.4%) reported at least one symptom. The most common symptoms were sore throat (n = 1229, 28.6%) and cough (n = 1154, 27.0%). The combination of fever with either cough or shortness of breath was reported by 6.9% of respondents (n = 295). Among those reporting any symptom, 2.6% (n = 46) reported having received testing. Among those reporting fever with either cough or shortness of breath, 5.7% (n = 15) reported having received COVID-19 testing.

More female than male respondents reported at least one symptom (45.3% vs 41.2%, p = 0.01, Table 2). Older persons (ages 65–74 and 75+) were less likely to report at least one symptom (p<0.0001) and the combination of fever with either cough or shortness of breath (p<0.0001). Indigenous/First Nations/Inuit/Metis had significantly higher rates of symptoms (49.3% vs 42.9%, p = 0.04) and testing (3.7% vs 1.1%, p = 0.0004) than those not reporting this background. This group (11.0% vs 6.5%, p = 0.005) and visible minorities (10.3% vs 6.3%, p = 0.001) also reported a higher rate of fever with cough or shortness of breath.

## COVID Near You- Canada-wide, April 3—April 26, 2020

After excluding duplicates, there were 409,207 responses. The median age was 42 years (IQR 33–54) and 58.0% (n = 237,150) were women (Table 1). Testing was reported in 0.2% (n = 612) of responses, and 0.4% (n = 1,479) reported seeing a health professional. Positive test results were reported in 0.03% (n = 105); some 0.1% (n = 213) reported that they were still waiting for their result. Among all respondents, 0.1% (n = 313) reported travel outside Canada in the previous two weeks and 0.1% (n = 324) reported contact with a known case of COVID-19.

The overall prevalence of symptoms was 1.6% (n = 6,746) and the most common symptoms were fatigue (n = 3,982, 1.0%), cough (n = 3,416, 0.8%) and headache (n = 3,406, 0.8%). The combination of fever with either cough or shortness of breath was reported by 0.2% of respondents (n = 758). Among those reporting any symptom, 8.9% (n = 598) reported being tested. Among those reporting fever with cough or shortness of breath, 21.0% (n = 159) reported being tested. Of the symptomatic who were tested, 17.2% (n = 103) reported a positive result.

More female than male respondents reported at least one symptom (2.0% vs 1.2%, p<0.001, Table 3), and were tested (0.2% vs 0.1%, p<0.001). Females and males had similar rates of positive test results (0.3% vs 0.2%, p = 0.44). Younger or middle-aged groups were more likely to report symptoms than older groups (p<0.001). Those under the age of 35 or over age 75 were more likely to have been tested (p = 0.009). A positive test result was significantly more common among those over age 75 (14% compared to 2–3% in other groups, p = 0.002). The rate of symptoms varied significantly across provinces–reporting at least one symptom was most common in British Columbia (2.1%) and Nova Scotia (2.0%, p<0.001) and reported testing rates were the highest in Nova Scotia (0.4%) and Saskatchewan (0.3%, p<0.001).

## Forum & Mainstreet Research phone poll- Ontario, April 11–12 and April 18–19, 2020

There were 9,147 unique households surveyed, and 41.7% (n = 4,165) consisted of at least 3 residents (Table 1). The survey respondents were more often women (53.3%, n = 4,931) than men. Completed testing was reported by 3.2% of all households (n = 299), and positive test results by 0.4% (n = 43). In addition, 0.5% (n = 50) were still awaiting test results.

The overall prevalence of any new symptom in the previous week was 14.9% (n = 1,385). The most common symptoms reported were headache (n = 662, 7.0%), sore throat (n = 377,

**Table 2. Prevalence of symptoms and testing within sociodemographic groups in Angus Reid poll, April 1–6, 2020[a].**

| | Any symptom, n (%) | Fever + (cough OR shortness of breath), n (%) | Reported testing, n (%) |
|---|---|---|---|
| **Age** | p<0.0001 | p<0.0001 | p = 0.72 |
| Under 35 years | 630 (52.0) | 113 (9.4) | 15 (1.4) |
| 35–54 | 701 (46.6) | 112 (7.2) | 23 (1.5) |
| 55–64 | 276 (36.4) | 40 (5.8) | 9 (1.2) |
| 65–74 | 197 (31.4) | 24 (3.6) | - |
| 75+ years | 59 (32.2) | 6 (3.3) | - |
| **Gender** | p = 0.02 | p = 0.14 | p = 0.03 |
| Female | 991 (45.3) | 159 (7.2) | 26 (1.2) |
| Male | 861 (41.2) | 133 (6.4) | 26 (1.2) |
| Other/no response | 11 (52.1) | - | - |
| **Age among Females** | p < 0.0001 | p = 0.04 | NA |
| Under 35 years | 335 (53.5) | 58 (8.8) | 8 (1.5) |
| 35–54 | 370 (49.1) | 63 (8.1) | 11 (1.5) |
| 55–64 | 148 (37.3) | 22 (6.4) | - |
| 65–74 | 106 (34.0) | 13 (4.2) | - |
| 75+ years | 32 (33.8) | - | - |
| **Age among Males** | p < 0.0001 | p = 0.003 | p = 0.94 |
| Under 35 years | 285 (50.0) | 52 (9.7) | 6 (1.1) |
| 35–54 | 331 (44.1) | 49 (6.2) | 12 (1.5) |
| 55–64 | 127 (35.3) | 18 (5.3) | - |
| 65–74 | 91 (28.3) | 11 (2.9) | - |
| 75+ years | 27 (30.6) | | - |
| **Annual Household Income (CAD)** | p = 0.36 | p = 0.54 | p = 0.26 |
| Under 25,000 | 197 (45.9) | 39 (8.7) | - |
| 25,000-<50,000 | 335 (43.7) | 50 (6.4) | 8 (1.1) |
| 50,000-<100,000 | 580 (44.4) | 97 (7.6) | 15 (1.3) |
| 100,000-<150,000 | 340 (43.1) | 52 (6.7) | 12 (1.4) |
| 150,000-<200,000 | 142 (45.6) | 15 (5.1) | 8 (2.7) |
| >200,000 | 65 (39.6) | 10 (6.7) | - |
| Don't know/would rather not say | 204 (38.9) | 32 (5.8) | - |
| **Highest Level of Education** | p = 0.13 | p = 0.80 | p = 0.99 |
| Secondary or less | 437 (40.6) | 75 (7.1) | 11 (1.3) |
| Some college or university | 903 (44.6) | 147 (7.1) | 25 (1.3) |
| Completed undergraduate | 374 (45.1) | 51 (6.4) | 11 (1.2) |
| Post-graduate degree | 149 (41.2) | 22 (5.8) | 6 (1.4) |
| **Indigenous/First nations/Inuit/Métis** | p = 0.04 | p = 0.005 | p = 0.0004 |
| | 161 (49.3) | 36 (11.0) | 11 (3.7) |
| **Visible minority** | p = 0.31 | p = 0.001 | p = 0.10 |
| | 245 (45.5) | 56 (10.3) | 10 (2.1) |
| **Province[b]** | p = 0.25 | p = 0.41 | NA |
| Alberta | 191 (44.6) | 33 (7.5) | - |
| British Columbia | 359 (45.7) | 54 (6.6) | 8 (1.2) |
| Manitoba | 124 (47.3) | 26 (10.7) | - |
| New Brunswick | 42 (51.5) | 8 (9.2) | - |
| Newfoundland/Labrador | 26 (36.1) | - | |
| Nova Scotia | 67 (46.8) | 7 (5.1) | - |
| Ontario | 499 (41.4) | 88 (7.3) | 13 (1.1) |

*(Continued)*

**Table 2.** (Continued)

| | Any symptom, n (%) | Fever + (cough OR shortness of breath), n (%) | Reported testing, n (%) |
|---|---|---|---|
| Quebec | 435 (43.1) | 59 (5.7) | 20 (2.0) |
| Saskatchewan | 115 (45.9) | 15 (6.1) | - |

[a] All percentage are weighted row percentages, reflecting the prevalence of column variables in each row group. p-values for between-group differences are at the top of each cell (for example in the top left cell, p-value is for the 5x2 table of age groups by any symptom yes/no). Cells <6 have been suppressed (denoted with a "-").

NA = not applicable (p-value could not be calculated due to zero cells and weighted data)

[b] Prince Edward Island results were suppressed due to small cells (< 6 observations).

**Table 3. Prevalence of self-reported symptoms, testing and positive test results within age, gender and province groups in COVID Near You poll, April 4–26, 2020[a].**

| | Any symptom, n (%) | Fever + (cough OR shortness of breath), n (%) | Reported testing, n (%) | Reported positive test result, n (%) |
|---|---|---|---|---|
| **Age** | *p <0.001* | *p = 0.44* | *p = 0.009* | *p = 0.002* |
| Under 35 years | 1,969 (1.7) | 227 (0.2) | 195 (0.2) | 31 (0.03) |
| 35–54 | 3,172 (1.6) | 348 (0.2) | 292 (0.1) | 45 (0.02) |
| 55–64 | 1,137 (1.8) | 121 (0.2) | 77 (0.1) | 13 (0.02) |
| 65–74 | 397 (1.3) | 49 (0.2) | 35 (0.1) | 9 (0.03) |
| 75+ years | 70 (1.4) | 13 (0.3) | 13 (0.3) | 7 (0.14) |
| **Gender** | *p <0.001* | *p <0.001* | *p <0.001* | *p = 0.003* |
| Female | 4,672 (2.0) | 511 (0.2) | 432 (0.2) | 61 (0.03) |
| Male | 1,904 (1.2) | 210 (0.1) | 158 (0.1) | 36 (0.02) |
| Other/no response | 170 (2.2) | 37 (0.5) | 22 (0.3) | 8 (0.11) |
| **Age among Females** | *p <0.001* | *p = 0.64* | *p = 0.34* | *p = 0.014* |
| Under 35 years | 1,335 (1.9) | 141 (0.2) | 132 (0.2) | 19 (0.03) |
| 35–54 | 2,229 (2.0) | 247 (0.2) | 216 (0.2) | 27 (0.02) |
| 55–64 | 807 (2.2) | 82 (0.2) | 55 (0.2) | 8 (0.02) |
| 65–74 | 271 (1.7) | 34 (0.2) | 24 (0.2) | - |
| 75+ years | 30 (1.5) | 7 (0.3) | - | - |
| **Age among Males** | *p < 0.001* | *p = 0.003* | *p = 0.003* | *p = 0.36* |
| Under 35 years | 562 (1.4) | 76 (0.2) | 54 (0.1) | 9 (0.02) |
| 35–54 | 870 (1.1) | 84 (0.1) | 68 (0.1) | 16 (0.02) |
| 55–64 | 320 (1.2) | 37 (0.1) | 19 (0.1) | - |
| 65–74 | 118 (0.9) | 10 (0.1) | 10 (0.1) | - |
| 75+ years | 34 (1.2) | 3 (0.1) | 7 (0.3) | - |
| **Province[b]** | *p < 0.001* | *p < 0.001* | *p < 0.001* | *p = 0.08* |
| Alberta | 868 (1.6) | 68 (0.1) | 97 (0.2) | 7 (0.01) |
| BC | 1483 (2.1) | 218 (0.3) | 95 (0.1) | 21 (0.03) |
| Manitoba | 242 (1.6) | 25 (0.2) | 16 (0.1) | - |
| New Brunswick | 91 (1.6) | 8 (0.1) | 9 (0.2) | - |
| Newfoundland /Labrador | 26 (1.5) | - | - | - |
| Nova Scotia | 269 (2.0) | 18 (0.1) | 49 (0.4) | - |
| Ontario | 3336 (1.6) | 377 (0.2) | 291 (0.1) | 67 (0.03) |
| PEI | 7 (1.2) | - | - | - |
| Quebec | 249 (1.2) | 22 (0.1) | 22 (0.1) | - |
| Saskatchewan | 170 (1.4) | 18 (0.2) | 31 (0.3) | - |

[a] All percentage are row percentages, reflecting the prevalence of column variables in each row group. p-values for between-group differences are at the top of each cell (for example in the top left cell, p-value is for the 5x2 table of age groups by "any symptom" yes/no). Cells <6 have been suppressed (denoted with a "-").

[b] Due to small cell sizes (<6), results for Yukon, Northwest Territories and Nunavut were suppressed.

**Table 4. Prevalence of self-reported symptoms, testing and positive test results within household groups in Forum & Mainstreet Research phone poll, April 11–12 and 18–19, 2020[a].**

| | Any symptom, n (%) | Fever + (cough OR shortness of breath), n (%) | Reported testing, n (%) | Reported positive test result, n (%) |
|---|---|---|---|---|
| **Household Income ($), n (%)** | *p = 0.002* | *p = 0.62* | *p = 0.17* | *p = 0.05* |
| Under 20,000 | 139 (16.2) | 12 (1.4) | 34 (4.2) | 10 (1.2) |
| 20,000-<60,000 | 411 (14.6) | 26 (0.8) | 93 (3.2) | 13 (0.5) |
| 60,000-<100,000 | 285 (14.1) | 15 (0.8) | 48 (2.4) | 7 (0.4) |
| >100,000 | 323 (17.0) | 15 (0.8) | 61 (3.1) | 7 (0.4) |
| Don't know/rather not say | 227 (12.7) | 14 (0.8) | 63 (3.5) | 6 (0.3) |
| **Household size, n (%)** | *p<0.0001* | *p = 0.005* | *p = 0.006* | *p = 0.28* |
| 1 | 202 (12.4) | 13 (0.8) | 44 (2.5) | 8 (0.4) |
| 2 | 454 (13.4) | 23 (0.1) | 100 (2.9) | 19 (0.5) |
| 3 | 236 (15.6) | 11 (0.2) | 42 (2.9) | - |
| 4 | 276 (18.2) | 9 (0.2) | 56 (3.6) | - |
| 5+ | 217 (19.6) | 26 (0.4) | 57 (5.0) | 9 (0.7) |

[a] All percentage are weighted row percentages, reflecting the prevalence of column variables in each row group. p-values for between-group differences are at the top of each cell (for example in the top left cell, p-value is for the 5x2 table of household income groups by any symptom yes/no). Cells <6 have been suppressed (denoted with a "-").

3.9%) and diarrhea (N = 345, 3.8%). The combination of fever with either cough or shortness of breath within the same household was reported by 0.8% (n = 82). Among those with any symptom, 6.5% (n = 94) reported that a household member had been tested. Among those with fever and either cough or shortness of breath, 37.5% (n = 31) reported that a household member had been tested. Positive test results were reported for 26.5% (n = 25) of all symptomatic households tested.

The lowest and highest income households had a significantly higher prevalence of COVID-19 symptoms (16.2% and 17.0%, p = 0.002, Table 4). The lowest income group was most likely to report a positive test result (1.2% in lowest vs 0.4% in highest, p = 0.05). The largest households were significantly more like to have at least one person with a COVID-19 symptom (19.6% in largest vs 12.4% in smallest, p<0.0001) and to report that at least one member was tested (5.0% vs 2.5%, p = 0.006). Households of one or 5+ persons were more likely to report flulike illness than households of 2–4 people (0.8% and 0.4% compared to 0.1–0.2%, p = 0.005).

## Discussion

In this study of syndromic surveillance data from three different survey sources, we find that described symptoms of COVID-19 were commonly reported by Canadian respondents. Specifically, 1.6% of respondents reported a symptom on the day of response, 15% of Ontario households had a new symptom in the previous week, and 43% of Canada-wide respondents had a symptom during March-early April 2020. Across the three studies, SARS-CoV-2-testing was reported in 2–9% of symptomatic responses, with a positive test rate among the symptomatic and tested of 17% in COVID Near You and 27% in the Forum Research poll. The three survey sources differed in geography (one covered only Ontario), time period (March to end of April 2020), and their representativeness across different demographic variables. Yet, after considering differences in the time window addressed with survey questions (present day, past week, past month), some consistent findings emerged.

In two different polls, women were more likely to report at least one symptom. In one poll, women were more likely to report testing. In Ontario, more women than men have been tested for SARS-CoV-2, yet men were more likely to have a positive test result [21]. Although the higher testing rate among women could reflect their greater presence in the healthcare sector, our findings also raise the possibility that women are more likely to report COVID-19-like symptoms. We further found that older respondents were less likely to report COVID-19 symptoms, but were more likely to test positive if tested. This higher self-reported rate of positivity is consistent with the concentration of early COVID-19 outbreaks among older Canadians, including (but not limited to) those residing in long-term care facilities (nursing homes) [22]. We found that Indigenous/First Nations/Inuit/Metis individuals reported a higher rate of symptoms and testing, and that visible minorities reported higher rates of fever with cough or shortness of breath. Residents of Indigenous communities were an early priority group for SARS-CoV-2 testing [5].

A report from the province of Ontario did not identify a consistent difference in testing rates across socioeconomic groups, although neighborhoods with higher ethnic concentration had a significantly higher rate of test positivity [23]. We did not identify significant differences in the frequency of possible COVID-19 symptoms across income or education groups at the level of the individual. However, we did find that households in the lowest income group were more likely to report symptoms and a positive test result among at least one resident. Larger households were also more likely to report that at least one person had symptoms or was tested–this may reflect the additional risk that comes from having more inhabitants or other characteristics potentially associated with larger households, such as level of education, income or ethnicity.

Whereas there were significant interprovincial differences in the proportion of COVID Near You respondents with symptoms, this was not the case for the Angus Reid poll. This may reflect differences in sample size, where a greater number of responses to COVID Near You meant that even small absolute differences in proportions reached statistical significance. Nonetheless, differences observed between provinces in both COVID Near You and the Angus Reid poll did not reflect differences in confirmed COVID-19 case activity. In COVID Near You, British Columbia and Nova Scotia had the highest proportion reporting at least one COVID-19 symptom. Yet, during March-April 2020, Quebec had considerably more cases than any other province [24]. This inconsistency with inter-provincial confirmed case trends likely reflects regional differences in survey uptake. Hence, some caution is warranted in attempting to compare rates of symptoms across provinces.

An important consideration in interpreting our findings is that many people with COVID-19 symptoms will not have COVID-19; conditions ranging from stress-related headaches and allergies to undiagnosed malignancies could also cause the same symptoms. Using only a more restrictive symptomatic definition such as fever with either cough or shortness of breath would miss many potential cases. Similarly, a recently developed algorithm that combines loss of smell or taste, fatigue, skipped meals, and cough, was only 65% sensitive for a positive SARS-CoV-2 test result [7]. To better understand current testing rates, we opted to use a broad symptom definition. This definition included anyone who would be eligible for testing on the basis of symptoms. To facilitate comparison, we also reported the proportion with fever and either cough or shortness of breath, an early syndromic definition used by the World Health Organization [20]. The weekly rate of household-level combination of fever with cough or shortness of breath in this study (Forum Research poll of Ontario in mid-April: 0.8%) was comparable to that obtained by the Public Health Agency of Canada's FluWatchers for the combination of cough and fever in early April 2020 (0.5%) [25].

There have been no previous reports of COVID-19 symptoms among the broader Canadian population published in the peer-reviewed literature. Our study provides essential information on the prevalence of such symptoms, and the proportion of symptomatic persons being tested. Strengths of this study are its inclusion of self-reported data from three distinct sources, covering March-April 2020. The consistency of our findings with published public health data suggests it is representative of the general population. Finally, the information we provide allows for a more complete picture of COVID-19 in Canada than just that which manifests through healthcare encounters. Lower barriers to diagnostic testing are essential given the growing understanding that COVID-19 can present with myriad symptoms. This will be helpful in identifying and isolating cases and preventing outbreaks as public health measures are lifted.

## Limitations

Our study also has several limitations. The variable time frames used in the three data sources complicate cross-study comparison, and longer time periods of self-report (e.g. "in the past month") may lead to higher levels of recall bias than shorter time periods. Similarly, household-level reporting does not easily compare to individual report, and combining symptoms experienced within a household may erroneously attribute all those symptoms to the same individual. Furthermore, survey questions varied in terms of symptoms covered and the inclusion of questions relating to healthcare encounters or testing results. Sample sizes were also quite small within subgroups, particularly when looking at those that reported testing or testing positive. Although the Angus Reid and Forum Research polls had a random sampling strategy, respondents on COVID Near You were self-selected, and so it was important to compare their characteristics, symptom reports, and testing rates to those obtained in the other two studies. Finally, despite their overall higher risk for COVID-19, those residing in long-term care and other institutional settings are likely not represented in these data sources which focus on community-dwelling residents of Canada.

## Conclusion

This study contributes essential data on the prevalence of COVID-19-related symptoms in Canada, and the proportion of symptomatic persons tested. This information complements public health-reported data on testing numbers and confirmed cases in Canada. We find that across three unique symptom surveys, less than 10% of those with symptoms in March-April 2020 reported having been tested for SARS-CoV-2. Our findings highlight the significant room to expand testing among community-dwelling residents of Canada. We have also identified groups with higher symptom prevalence (women, younger age groups, Indigenous/First Nations/Inuit/Métis), information which can be used to refine testing strategies and guide outreach efforts. Syndromic surveillance data such as these can supplement public health reports and provide much-needed context to gauge the adequacy of current SARS-CoV-2 testing rates.

## Supporting information

**S1 Table. Survey response rates.**
(DOCX)

**S2 Table. Angus Reid poll questions used in this study.**
(DOCX)

**S3 Table. COVID Near You tool questions used in this study.**
(DOCX)

**S4 Table. Forum poll questions used in this study.**
(DOCX)

## Acknowledgments

We thank Boston Children's Hospital, the Angus Reid Institute, Forum Research and Mainstreet Research for providing the data used in this study. Boston Children's Hospital, the Angus Reid Institute, Forum Research and Mainstreet Research bear no responsibility for the analyses or interpretations of the data presented here.

## Author Contributions

**Conceptualization:** Jared B. Hawkins, Todd C. Lee, Jessica J. Liu, Noah M. Ivers, Nathan M. Stall, Effie Gournis, Isaac I. Bogoch, John S. Brownstein.

**Data curation:** Lauren Lapointe-Shaw, Benjamin Rader, Christina M. Astley, Jared B. Hawkins, William J. Schatten.

**Formal analysis:** Lauren Lapointe-Shaw, Benjamin Rader, Christina M. Astley, Deepit Bhatia.

**Funding acquisition:** Lauren Lapointe-Shaw.

**Investigation:** Lauren Lapointe-Shaw.

**Methodology:** Lauren Lapointe-Shaw, Christina M. Astley, Noah M. Ivers, Ashleigh R. Tuite, David N. Fisman, Isaac I. Bogoch, John S. Brownstein.

**Project administration:** Lauren Lapointe-Shaw, Isaac I. Bogoch, John S. Brownstein.

**Resources:** Isaac I. Bogoch, John S. Brownstein.

**Writing – original draft:** Lauren Lapointe-Shaw.

**Writing – review & editing:** Lauren Lapointe-Shaw, Benjamin Rader, Christina M. Astley, Jared B. Hawkins, Deepit Bhatia, William J. Schatten, Todd C. Lee, Jessica J. Liu, Noah M. Ivers, Nathan M. Stall, Effie Gournis, Ashleigh R. Tuite, David N. Fisman, Isaac I. Bogoch, John S. Brownstein.

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
