## [Decision Letter · Decision Letter 0]

22 Jul 2020

PONE-D-20-18024

Web and Phone-based COVID-19 Syndromic Surveillance in Canada: A Cross-Sectional Study

PLOS ONE

Dear Dr. Lauren Lapointe-Shaw

Thank you for submitting your manuscript to PLOS ONE. After careful consideration, we feel that it has merit but does not fully meet PLOS ONE’s publication criteria as it currently stands. Therefore, we invite you to submit a revised version of the manuscript that addresses the points raised during the review process.

We look forward to receiving your revised manuscript.

Kind regards,

Francesco Di Gennaro

Academic Editor

PLOS ONE

Journal Requirements:

2. In ethics statement in the manuscript and in the online submission form, please provide additional information about the patient records used in your retrospective study. Specifically, please ensure that you have discussed whether all data were fully anonymized before you accessed them and/or whether the IRB or ethics committee waived the requirement for informed consent. If patients provided informed written consent to have data from their medical records used in research, please include this information.

3. Thank you for including your competing interests statement; "I have read the journal's policy and the authors of this manuscript have the following competing interests: WJ Schatten is a paid employee for Forum Research. II Bogoch has consulted to BlueDot, a social benefit corporation that tracks the spread of emerging infectious diseases. The remaining authors have no disclosures."

We note that one or more of the authors are employed by a commercial company: Forum Research

4. We note you have included a table to which you do not refer in the text of your manuscript. Please ensure that you refer to Table 3 in your text; if accepted, production will need this reference to link the reader to the Table.

Additional Editor Comments (if provided):

I appreciate your paper, but need some revisions.

Following reviewer suggestions you can improve your manuscript

Reviewers' comments:

Reviewer's Responses to Questions

**Comments to the Author**

1. Is the manuscript technically sound, and do the data support the conclusions?

Reviewer #1: Yes

Reviewer #2: Partly

2. Has the statistical analysis been performed appropriately and rigorously? 

Reviewer #1: Yes

Reviewer #2: I Don't Know

3. Have the authors made all data underlying the findings in their manuscript fully available?

Reviewer #1: Yes

Reviewer #2: No

4. Is the manuscript presented in an intelligible fashion and written in standard English?

Reviewer #1: Yes

Reviewer #2: Yes

5. Review Comments to the Author

Reviewer #1: This article provides estimates of the prevalence of COVID-19 symptoms in Canada which is valuable for modelling and public health planning. A few points for consideration:

1) How are the Angus Reid Forum panel members selected? Is this a random sample of Canadians?

2) At the start of page 8, Table 2 is referenced and I believe this should be Table 3.

3) Verify data in Table 3 as the proportion reporting at least one symptom for Ontario on page 8 differs from what is in Table 3

4) No regional differences were observed with the Angus Reid data but regional differences were observed in the COVID near you data. Any reasons to explain why?

Reviewer #2: Title: Web and Phone-based COVID-19 Syndromic Surveillance in Canada: A Cross-Sectional Study

This paper presents a study to describe the characteristics, symptoms, and self-reported testing rates of respondents in three different COVID-19 symptom surveys in Canada. However, there are questions that limit my enthusiasm of the paper, as outlined below.

1. Results:

a. Table 1: Authors considered (-) and 0. What I guess (-) shows cell<6. So please define that at the caption and fix that across all 3 tables. We don’t expect to have both 0 and (-) across tables.

b. Table 2: Authors did stratification for the age based on the gender.

i. Did authors find gender as a cofounder or important variable that is associated with Fever + (cough or shortness of breath)/Any symptom? Please clarify this part. In other words, I would like to know the reason of stratification of age by gender.

ii. At least for COVID Near You, there is enough samples for other/no response group. Please modify Table 3 and add that group result to the Table.

iii. Why not to consider age as an individual variable without being classified by gender and be added to the Tables. How about adding gender (F/M/other) to the Tables as well?

iv. Tables 2 and 3 can’t be followed easily. Please modify the tables.

v. (Rao-Scott) Chi-squared/Fisher tests assess the association between two categorical variables, or comparing proportions across cells for a given variable. Authors considered these methods to compare the proportions of cells (e.g., age groups) for a given variable (e.g., any symptom), is it right?

vi. Is there any reported testing results for Angus Reid Poll study (Table 2)?

c. Why authors didn’t include Table for Forum and mainstreet research phone poll? Please clarify this part.

2. I suggest authors consider parametric methods (e.g., logistic regression model) to add more results regarding the association between the demographic variables and two main variables (1) Fever + (cough OR shortness of breath) vs other symptom and (2) Reported positive test result/not.

3. In addition to the previous comment, how about comparing the results between web and phone-based sources? Authors introduced these three data sources, however there is not enough results to compare the data across these three studies.

4. Authors should be more precise about calling Tables across manuscript. Page 8, line 1 (results related to the COVID Near You), shows Table 3, not Table 2.

5. To improve results due to the lack of samples, integrating these three studies using meta-analysis approaches may improve results and power of analysis.

6. PLOS authors have the option to publish the peer review history of their article (what does this mean?). If published, this will include your full peer review and any attached files.

Reviewer #1: No

Reviewer #2: No

---

## [Author Response · Author response to Decision Letter 0]

19 Aug 2020

Please see submitted "responses" document.

---

## [Decision Letter · Decision Letter 1]

16 Sep 2020

Web and phone-based COVID-19 syndromic surveillance in Canada: a cross-sectional study

PONE-D-20-18024R1

Dear Dr. Lapaoint-Shaw,

We’re pleased to inform you that your manuscript has been judged scientifically suitable for publication and will be formally accepted for publication once it meets all outstanding technical requirements.

Kind regards,

Francesco Di Gennaro

Academic Editor

PLOS ONE

Additional Editor Comments (optional):

Dear Authors, congratulations!

Reviewers' comments:

Reviewer's Responses to Questions

**Comments to the Author**

1. If the authors have adequately addressed your comments raised in a previous round of review and you feel that this manuscript is now acceptable for publication, you may indicate that here to bypass the “Comments to the Author” section, enter your conflict of interest statement in the “Confidential to Editor” section, and submit your "Accept" recommendation.

Reviewer #1: All comments have been addressed

Reviewer #2: All comments have been addressed

2. Is the manuscript technically sound, and do the data support the conclusions?

Reviewer #1: Yes

Reviewer #2: Yes

3. Has the statistical analysis been performed appropriately and rigorously? 

Reviewer #1: Yes

Reviewer #2: Yes

4. Have the authors made all data underlying the findings in their manuscript fully available?

Reviewer #1: No

Reviewer #2: Yes

5. Is the manuscript presented in an intelligible fashion and written in standard English?

Reviewer #1: Yes

Reviewer #2: Yes

6. Review Comments to the Author

Reviewer #1: I have reviewed the responses to my questions and concerns. They have been addressed sufficiently. I have no further comments. I note that there are some restrictions to making the data publicly available and they appear to be warranted.

Reviewer #2: All the comments have been addressed. Just a minor comment is related to introduce the IQR in the method section before using at result section. Thank you

7. PLOS authors have the option to publish the peer review history of their article (what does this mean?). If published, this will include your full peer review and any attached files.

Reviewer #1: No

Reviewer #2: No

---

## [Editor Report · Acceptance letter]

25 Sep 2020

PONE-D-20-18024R1 

Web and phone-based COVID-19 syndromic surveillance in Canada: a cross-sectional study 

Dear Dr. Lapointe-Shaw:

I'm pleased to inform you that your manuscript has been deemed suitable for publication in PLOS ONE. Congratulations! Your manuscript is now with our production department. 

Kind regards, 

on behalf of

Dr. Francesco Di Gennaro 

Academic Editor

PLOS ONE